# Identification of *CDPK* Gene Family in *Solanum habrochaites* and Its Function Analysis under Stress

**DOI:** 10.3390/ijms23084227

**Published:** 2022-04-11

**Authors:** Yingying Li, Haixin Zhang, Sibo Liang, Xiuling Chen, Jiayin Liu, Yao Zhang, Aoxue Wang

**Affiliations:** 1College of Life Sciences, Northeast Agricultural University, Harbin 150030, China; l18645066071@163.com (Y.L.); z919843847@163.com (H.Z.); 2College of Horticulture and Landscape Architecture, Northeast Agricultural University, Harbin 150030, China; 13100956039@163.com (S.L.); chenx@neau.edu.cn (X.C.); 3College of Sciences, Northeast Agricultural University, Harbin 150030, China; 13040216@163.com

**Keywords:** bioinformatics, *CDPK* gene family, *Solanum habrochaites*, *ShCDPK6* and *ShCDPK26*, stress

## Abstract

Tomato is an important vegetable crop. In the process of tomato production, it will encounter abiotic stress, such as low temperature, drought, and high salt, and biotic stress, such as pathogen infection, which will seriously affect the yield of tomato. Calcium-dependent protein kinase (CDPK) is a class of major calcium signal receptor which has an important regulatory effect on the perception and decoding of calcium signals. CDPK plays a key role in many aspects of plant growth, such as the elongation of pollen tubes, plant growth, and response to biotic and abiotic stress. While some studies have concentrated on *Arabidopsis* and pepper, *Solanum habrochaites* is a wild species relative of cultivated tomato and there is no report on CDPK in *Solanum habrochaites* to date. Using tomato genomic data, this study identified 33 members of the *CDPK* gene family. Evolutionary analysis divides family members into four Asian groups, of which the CDPK family members have 11 gene replication pairs. Subcellular location analysis showed that most proteins were predicted to be located in the cytoplasm, and less protein existed on the cell membrane. Not all CDPK family members have a transmembrane domain. Cis regulatory elements relating to light, hormones, and drought stress are overrepresented in the promoter region of the *CDPK* genes in *Solanum habrochaites*. The expression levels of each gene under biotic stress and abiotic stress were quantified by qRT-PCR. The results showed that members of the CDPK family in *Solanum habrochaites* respond to different biotic and abiotic stresses. Among them, the expression of *S**hCDPK6* and *S**hCDPK26* genes change significantly. *S**hCDPK6* and *S**hCDPK26* genes were silenced using VIGS (virus-induced gene silencing), and the silenced plants illustrated reduced stress resistance to *Botrytis cinerea,* cold, and drought stress. The results of this study will provide a basis for the in-depth study of the *CDPK* gene family in *Solanum habrochaites*, laying the foundation for further analysis of the function of the gene family.

## 1. Introduction

Plants often encounter a wide variety of abiotic and biotic stresses in nature and thus they have evolved a complex defense system to resists external pressure. This system facilitates the recognition and perception of external attack, which in turn will be converted to defense mode. Ca^2+^ is an important second messenger in the transduction of plant cell signaling [1]. When plants are subjected to an external adverse stress, calcium signals in cells are transmitted to downstream components through calcium sensing proteins, such as calmodulin, Calcinerium-B Like, and the Calcium-dependent protein kinase (CDPK) proteins, thereby modulating the expression of associated genes [2,3,4]. In plants, cytoplasmic regulation of Ca^2+^ concentration can be a response to various endogenous and exogenous signals, including changes in hormone level, abiotic stress (such as drought, high and low temperature, or light), and biotic stress (such as pathogens and non-pathogenic microbes) [5,6,7,8,9]. External signals lead to a short increase in Ca^2+^ concentration in the cytoplasm. Ca^2+^ is combined with the calcium binding protein (CBP) or Ca^2+^ sensor, transmitting signals and causing cells to change during biochemical and physiological processes [10,11]. In addition, Ca^2+^ also help maintain cell wall and cell membrane stability through the regulation of physiological processes, such as stomatal guard cell movement, root hair elongation, and pollen tube growth [12,13]. There are five types of calcium ion sensors in plants, including calcium modes (CAM), calcium modulous protein (CML), calcineriumB protein (CBL), calcium/calcium modulus protein kinase (CAMK), and calcium-dependent protein kinase (CDPK) [14,15]. Compared to CAM, CML, and CBL, CDPKs direct Ca^2+^ signals to the phosphorylation level. This confers CDPKs with dual functions, namely Ca^2+^ sensor and responder [16].

CDPKs are widely distributed in plants [3]. Specific expression of *CDPK* genes can be detected in different tissues, including roots, stems, leaves, and flowers, among others [17,18,19]. The *CDPK* gene is generally present in a branch tissue cell, a wooden portion cell, a pollen mother, and embryonic cells [20]. The wide distribution of CDPKs among many plant species is an indication of the importance of these proteins in signal transduction pathways. For a long time, researchers have thought that CDPK exists in plants. Genomic analysis revealed that there are 34 CDPKs in *Arabidopsis,* 31 in rice, 20 in wheat, 20 in poplar, and 40 in corn [3,20]. The *CDPK* family consists of six protein kinases, including calcium-dependent protein kinase (CDPKs), CDPK-related kinase (CRKs), phosphate kettate carboxylase kinase (PPCKs), phosphate ketone carboxylase kinase-related kinase (PEPRKS), calcium mode in regulatory kinase (CAMKs), and calmodulin dependent protein kinase (CDPKs). The main difference between them is that they have different regulatory domains. CDPK has a conservative structure including a variable N-terminal domain, a Ser/Thr protein kinase domain, and an automatic suppression structure, usually included four EF-hands structures [21].

Many members of the *CDPK* family play important roles in plant response to abiotic stress and in disease resistance. OsCDPK1 regulates rice salt and drought tolerance [22]. The *CDPK* family is involved in drought or salt stress adaptation by inducing ABA responsive genes and adjusting the ABA-induced anionic channel (SLAC1, SLAH3), leading to stomata adjustment in *Arabidopsis* [23,24]. Overexpression of *MDCPK1A* in tobacco removes ROS accumulation and regulates the expression of stress-related genes, thereby significantly increasing cold and salt resistance [25]. Excessive expression of *ZMCPK1* in corn leaf inhibits the expression of cold inducing marker gene *ZMERF3*. The ectopic expression of *ZMCPK1* in *Arabidopsis* reduced the adaptation of the plant in terms of cold resistance, indicating that ZMCPK1 acts as a negative regulatory factor under cold stress [26]. *Arabidopsis* CPK28 functions as a negative regulator of an immune signal which responds to immune responses by regulating BIK1 (multi-mode identification receptor (PRR)) [3]. CDPK can also be used as a negative regulator of the stress response. Transgenic plants overexpressing *CDPK* are more sensitive to abiotic stress and biotic stress. *Arabidopsis cpk23* mutant increases tolerance to drought and salt stress, but resistance to drought and salt stress is reduced in *AtCPK23* overexpressing plants [27]. Taken together, it is apparent that CDPKs participate in plant abiotic and biotic stress in the form of positive and negative regulation. Virus-induced silencing of *CDPK2* and *CDPK3* showed that CDPK1 and CDPK2 are involved in the regulation of *AVR9/CF-9* genes, and CDPK is a necessary condition for plant AVR9/CF-9 induced hypersensitivity during pathogen infection [28]. CDPK10 in maize is also involved in defense signaling pathways [29].

As an important economic crop, tomatoes often suffer from biotic and abiotic stress, such as drought, salt, and low temperatures. At the same time, as a widely cultivated vegetable crop and model plant, tomato plays an important role in vegetables and plant research. *Solanum habrochaites* is the nearest wild relative of cultivated tomato and has strong resistance to cold stress [30,31,32,33]. It is an excellent germplasm material for the agronomic improvement of cultivated tomato. *S. habrochaites* can cross with cultivated tomato only when it is used as male parent, but the fruit seed setting rate is low and it is difficult to cross pests. It is difficult to transfer the excellent characteristics of *S. habrochaites* to cultivated varieties. Therefore, as superior stress resistant materials, *Solanum habrochaites* often play an important role in improving the quality characteristics of tomatoes.

While CDPKs’ important role in calcium signal transduction has been studied in *Arabidopsis* and corn, among others, it has not been studied in *Solanum habrochaites*. We suspect that the *CDPK* gene family plays an important role in affecting the strong low temperature and disease resistance of *Solanum habrochaites*. In order to analyze the function and genetic evolution of the *CDPK* gene family in *Solanum habrochaites*, the genomic data of *Solanum habrochaites* were first studied by bioinformatics, where the gene family members of *CDPKs* were identified and their gene and protein sequences, conservative motif, and cis acting elements were analyzed in this study. The expression of the *CDPK* gene family under biotic stress (*Botrytis cinerea*) and abiotic stress, such as low temperature and drought, was investigated by qRT-PCR. The functions of *ShCDPK6* and *ShCDPK26* genes were further verified by virus-induced gene silencing. This study provides a theoretical basis for clarifying the biological function of *CDPK* genes in response to stress, and further lays a foundation for tomato crop improvement.

## 2. Results

### 2.1. Identification of CDPK Family Members in Solanum habrochaites

In order to identify the members of the *C**DPK* family in *Solanum habrochaites*, we conducted local tBLASTn comparison between the nucleotide sequences of the existing tomato annotated genes and the CDPK amino acid sequences of *Arabidopsis thaliana* (E-value < 1 × 10^−7^, identity >50%), removing duplicates to obtain tomato candidate CDPK. We used the Prosite (http://prosite.expasy.org/ accessed on 24 July 2020) website to identify candidate CDPK protein domains, screening CDPKs with NAF domain structure, and eventually received 33 members of the *C**DPK* family (Table 1). These genes were named *ShCDPK1–ShCDPK33* according to chromosome location information. The *CDPK* gene family is distributed on 12 chromosomes, of which six family members are distributed on chromosome 1. We analyzed the physical and chemical properties of *CDPK* family gene length, amino acid length, isoelectric point, and molecular weight of *Solanum habrochaites*. It was found that the gene length of *CDPK* family members varied greatly, ranging from 1506 to 3204 bp, of which *ShCDPK19* gene was the shortest and *ShCDPK31* gene was the longest. The range of amino acids encoded by CDPK family is 501~1067 aa. The minimum isoelectric point (ShCDPK17) was 4.77 and the maximum isoelectric point (ShCDPK8) was 9.77. The members of the gene family were acid protein and basic protein. Subcellular localization prediction analysis showed that most genes were localized in the cytoplasm and two family members existed not only in the cytoplasm, but also on the cell membrane. All family members predicted that there was no transmembrane domain (Table 1).

### 2.2. Phylogenetic Analysis of CDPK Gene in Solanum habrochaites

In order to further understand the evolutionary relationship of *CDPK* gene in *Solanum habrochaites*, we constructed a phylogenetic tree with *Arabidopsis*, tomato and pepper CDPK family members. It can be seen from Figure 1 that 98 *CDPK* gene evolutionary trees are divided into four subfamilies, named 1~4 respectively. One subgroup includes family members ShCDPK1, ShCDPK5, ShCDPK6, ShCDPK14, ShCDPK15, ShCDPK17, ShCDPK19, ShCDPK23, ShCDPK24, ShCDPK26, ShCDPK27, ShCDPK30, and ShCDPK31. Parallel homologous pairs include ShCDPK6 and ShCDPK24, ShCDPK14 and ShCDPK31, ShCDPK19 and ShCDPK30. Vertical homologous pairs include ShCDPK1 and CaCDPK6, ShCDPK5 and CaCDPK22, ShCDPK17 and CaCDPK19. 2, while subgroups include: ShCDPK3, ShCDPK13, ShCDPK16, ShCDPK20, ShCDPK21, ShCDPK29, ShCDPK32, and ShCDPK33. Parallel homologous pairs include ShCDPK3 and ShCDPK29, ShCDPK20 and ShCDPK33. Vertical homologous pairs ShCDPK13 and CaCDPK7, ShCDPK16 and AtCPK9, and ShCDPK21 and AtCPK3. Three subgroups include ShCDPK4, ShCDPK10, ShCDPK11, ShCDPK18, ShCDPK22, ShCDPK25, and ShCDPK28. Parallel homologous pairs include ShCDPK4 and ShCDPK25, ShCDPK10 and ShCDPK11. Vertical homologous pairs include ShCDPK18 and AtCPK24, ShCDPK22 and AtCPK13. Four subgroups include ShCDPK2, ShCDPK7, ShCDPK8, ShCDPK9, and ShCDPK12. The results showed that the *CDPK* gene family was similar between *Solanum habrochaites* and cultivated tomato (Figure 1).

### 2.3. Analysis of CDPK Gene Family Structure in Solanum habrochaites

The gene structural analysis by GSDs tool show that the *CDPK* gene family in *Solanum habrochaites* was mainly enriched by introns. *ShCDPK8* and *ShCDPK12* have 11 introns. *ShCDPK2, ShCDPK7*, and *ShCDPK9* have 10 introns. *ShCDPK25* has nine introns. *ShCDPK4, ShCDPK13, ShCDPK18, ShCDPK27, ShCDPK28, ShCDPK31*, and *ShCDPK32* have eight introns. *ShCDPK1, ShCDPK6, ShCDPK16, ShCDPK19, ShCDPK20, ShCDPK21, ShCDPK23, ShCDPK24, ShCDPK29*, and *ShCDPK33* have seven introns. *ShCDPK3, ShCDPK5, ShCDPK10, ShCDPK11, ShCDPK14, ShCDPK15, ShCDPK17, ShCDPK22, ShCDPK26,* and *ShCDPK30* have six introns. The above members belong to the intron enrichment group (Figure 2).

### 2.4. Motif Analysis of CDPK Gene Family in Solanum habrochaites

Ten motifs were predicted in the tomato CDPK family by meme software, and their motif types and sequence were basically the same. The higher the homology, the stronger the similarity of gene motif arrangement (Figure 3). Hence, 33 *ShCDPK* genes have the same conservative motif and order. *ShCDPK4, ShCDPK10, ShCDPK11, ShCDPK18, ShCDPK22, ShCDPK25,* and *ShCDPK28* do not have motif 9. Compared with other family members, they contain a unique motif 10. *ShCDPK4, ShCDPK10, ShCDPK11, ShCDPK18*, and *ShCDPK22* do not have motif 4 or motif 10, but have two motif 6.

### 2.5. Gene Replication and Collinearity Analysis of CDPK Gene Family in Solanum habrochaites

The gene replication of *CDPK* gene family in *Solanum habrochaites* is analyzed in Appendix A. *ShCDPK1:ShCDPK27**, ShCDPK4:ShCDPK28**, ShCDPK5:ShCDPK23**, ShCDPK6:ShCDPK24**, ShCDPK6:ShCDPK26**, ShCDPK24:ShCDPK26**, ShCDPK30:ShCDPK19**, ShCDPK20:ShCDPK33**, ShCDPK7:ShCDPK9**, ShCDPK8:ShCDPK12*, and *ShCDPK15:ShCDPK17* have 11 pairs of gene replication pairs. The KS range of four repetitions is 0.0932819–1.30348, and the separation time range can be inferred to be 8.2–115.2 Mya. Except for *ShCDPK7* and *ShCDPK9*, the repeated Ka/Ks values of the fragments of the 10 replication gene pairs are less than 1, indicating that they have been purified and selected (Appendix A).

### 2.6. Analysis of Cis Acting Elements of CDPK Gene Family in Solanum habrochaites

The structure of the promoter affects the affinity between promoter and RNA polymerase, thus affecting the level of gene expression. By analyzing the 1500 bp upstream of *CDPK* gene family members in *Solanum habrochaites*, many cis acting elements associated with plant stress were determined (Appendix A). In the analysis of cis acting elements of *ShCDPKs*, in addition to the common CAAT box and TATA box originals, they can be divided into three categories. The first category comprises hormone related components, such as ABRE and CGTCA motif. The second category is the original related to regulation, such as light regulation. The third category is the original related to biological stress, such as TC rich repeats, MBS, and LTR. All members have photoregulated cis acting elements, 23 family members have abscisic acid (ABRE), 19 family members contain methyl jasmonate (CGTCA motif), 19 family members have cis acting elements of defense and adversity responses (TC rich repeats), and 18 family members contain anaerobic induction cis acting elements (ARE), 13 family members have salicylic acid cis acting elements (TCA), 11 family members have cold stress cis acting elements (LTR), nine family members have drought stress cis acting elements (MBS), 10 family members have auxin related cis acting elements (TGA element), and six family members have gibberellin related cis acting elements(P-Box).

### 2.7. Expression Analysis of CDPK Gene Family in Solanum habrochaites under Various Stress

Bioinformatics analysis revealed that members of the *CDPK* gene family in *Solanum habrochaites* were associated with multiple stresses (biotic and abiotic). Therefore, in order to further analyze the function of the *CDPK* gene family, we treated tomato seedlings with low temperature (4 °C), drought, and *Botrytis cinerea* and detected the gene expression under stress by qRT-PCR.

The expression of the *CDPK* gene family in *Solanum habrochaites* changed to a certain extent under cold stress. The expression of *ShCDPK5, ShCDPK6, ShCDPK1**2, ShCDPK**15, ShCDPK2**6, ShCDPK29,* and *ShCDPK33* changed significantly under cold stress. The expressions of *ShCDPK5* and *ShCDPK6* were significantly up-regulated after cold induction for 0.5, 12, and 24 h. The expression of *ShCDPK5* at 12 h was about 11.0 times that of untreated leaves (the expression at 0 h, CK). The expression of *ShCDPK6* was the highest at 12 h, about 31.8 times that of the control group (CK). *ShCDPK19**, ShCDPK29,* and *ShCDPK33* showed an upward trend as a whole. The highest value of *ShCDPK19* appeared at 24 h, about 18.1 times that of the control group. The highest value of the *ShCDPK29* occurred at 12 h and was about 12.7 times higher than the control group. The highest value of the *ShCDPK**33* occurred at 12 h and was about 26.1 times higher than the control group. The expression of *ShCDPK26* increased significantly at the beginning of stress, and the highest expression was found at 12 h, about 38.9 times that of the control group. In conclusion, *ShCDPK6* and *ShCDPK26* were the most affected by cold stress in the *CDPK* gene family of *Solanum habrochaites*, followed by *ShCDPK19* (Figure 4).

Under drought induction, the expression of *CDPK* gene family in *Solanum habrochaites* changed visibly, and the most significant appeared at 0.5 h. At this time, those with high relative expression include *ShCDPK6, ShCDPK23,* and *ShCDPK26,* and the expression of these genes can reach levels higher than 30 times. Most genes showed a wave expression trend and peaked at different time points. The expression of *ShCDPK6* gene was 32.5 times that of the control group at 0.5 h and 16.4 times that of the control group at 12 h. The expression of *ShCDPK23* gene was 31.0 times that of the control group at 0.5 h and 17.9 times that of the control group at 12 h. The expression of *ShCDPK26* gene was 32.9 times that of the control group at 0.5 h and 20.7 times that of the control group at 12 h. *ShCDPK6, ShCDPK8, ShCDPK10, ShCDPK16, ShCDPK18, ShCDPK20, ShCDPK21, ShCDPK23, ShCDPK26, ShCDPK29, ShCDPK30,* and *ShCDPK33* genes was higher than that of 0 h at all time points, with the most pronounced expression at 0.5 h and 24 h. In conclusion, after drought treatment, several genes with significant gene changes are *ShCDPK6, ShCDPK16, ShCDPK18, ShCDPK23, ShCDPK26, ShCDPK29, ShCDPK30,* and *ShCDPK33* (Figure 5).

We detected the expression changes of the *CDPK* gene family under *Botrytis cinerea* stress. Through qRT-PCR detection, we found that the expression of the *CDPK* gene family in *Solanum habrochaites* changed in varying degrees under *Botrytis cinerea* stress, and most of them changed significantly in 0.5 h. *ShCDPK4*, *ShCDPK12*, *ShCDPK20,* and *ShCDPK22* genes had the same expression change trend, and showed high expression at 0.5 h, with little change at other points in time. The expression of *ShCDPK6*, *ShCDPK18*, *ShCDPK19*, *ShCDPK25*, *ShCDPK26*, *ShCDPK29*, *ShCDPK30*, *ShCDPK31,* and *ShCDPK33* decreased at 0.5–9 h and increased at the later stage. The expression of *ShCDPK6* and *ShCDPK33* was overtly raised in the later stage of stress, and at 36 h, the expression of both genes was 43.9 times and 32.8 times higher than that of the control. The peak values of *ShCDPK26*, *ShCDPK29,* and *ShCDPK31* were 25.6 times, 15.5 times, and 14.9 times that of the control at 36 h, respectively. In conclusion, *ShCDPK6* and *ShCDPK33* were the most affected by *Botrytis cinerea* stress in the *CDPK* gene family of *Solanum habrochaites*, followed by *ShCDPK26* (Figure 6).

### 2.8. Functional Analysis of Silencing ShCDPK6 and ShCDPK26 in Solanum habrochaites

Gene expression differences between *Sh**CDPK6* and *Sh**CDPK26* were the most obvious after three stress treatments. It is speculated that they may be involved in the response of *Solanum habrochaites* to stress. We will take *ShCDPK6* and *ShCDPK26* as the research objects to further verify the function of the gene through VIGS. VIGS vectors *PTRV2-ShCDPK6* and *PTRV2*-*ShCDPK26* were successfully constructed and transformed into *Agrobacterium* *tumefaciens*. Tomato seedlings were injected with *agrobacterium* *tumefaciens* containing the target vector and PDS (Phytoene dehydrogenase) silenced plants were used as indicators of the VIGS silencing effect. The expression levels of target genes in *ShCDPK6* and *ShCDPK26* silent plants and control plants were detected by fluorescence quantitative PCR, and the silencing efficiency of *ShCDPK6* and *ShCDPK26* is 69.4% and 48.5% respectively (Appendix A). Finally, we obtained 15 *ShCDPK6* and *ShCDPK26* silencing plants respectively. In terms of bioinformatics analysis, *ShCDPK6, ShCDPK24*, and *ShCDPK26* genes were found to have a close evolutionary relationship, and the expression of *ShCDPK24* gene was analyzed separately for the silenced plants, as shown in Appendix A. The results revealed that the expression of *ShCDPK24* gene in *ShCDPK6* and *ShCDPK26*-silenced plants did not show significant changes in expression.

Plants will produce some reactive oxygen species after encountering adversity, which will have a toxic effect on cells, leading to plant cell and tissue damage and death. Therefore, under *Botrytis cinerea*, drought, and cold stress, various physiological indexes of *ShCDPK6* and *ShCDPK26* silenced plants and control plants were measured respectively to evaluate the degree of oxidative damage between them. The contents of superoxide dismutase (SOD), catalase (CAT), ascorbate oxidase (AAO), peroxidase (POD), and polyphenol oxidase (PPO) in silenced and control plants were determined under low temperature (4 °C), drought, and biotic stress (*Botrytis cinerea*). Under cold stress, SOD, AAO, POD, and PPO values of silenced plants were lower than those of control plants, indicating that the cold resistance of the silenced plants was reduced (Figure 7). Under drought treatment, the contents of CAT, POD, and PPO in silenced plant were significantly lower than that of control group. The change in the dynamic of PPO content was in keeping with that of the control plant in the early stage of stress. *ShCDPK6* and *ShCDPK26* decreased in varying degrees after 9 h (Figure 8). Under *Botrytis cinerea* treatment, changes of SOD, POD, PPO, and AAO contents in the *ShCDPK6* silenced plants was distinctly lower than that of the control group, while the changes of CAT, POD, and PPO in silent plant *ShDPK26* were smaller than those in control group, indicating that the *Botrytis cinerea* resistance of the silenced plants was reduced (Figure 9). The phenotype of the control and silenced plants under three stresses is shown in Appendix A. Compared with the control plants, the silent plants showed more wilting under low temperature and drought stress, but the phenotypic difference was not very obvious under *Botrytis cinerea* stress.

## 3. Discussion

The *CDPK* gene family widely exists in all parts of plants, such as roots, stems, leaves, flowers, fruits, and seeds, and plays an important role in plant state construction [17,34]. *Arabidopsis* CPK11 positively regulates the ABA signaling pathway in seed germination and stomatal movement [35]. Later, it was found that *Arabidopsis* CPK12, as a homologue of CPK4 and CPK11, negatively regulates the process of plant seed germination and growth after germination and antagonizes CPK4 and CPK11 in the ABA signaling pathway [34]. CDPKs also play an important role in plant growth and development and response to biotic and abiotic stresses. This has been identified and studied in many species [36]. For example, the first purification and identification were carried out in soybean. Cucumber CsCDPK5 is involved in the formation of adventitious roots of hypocotyl [37]. *Arabidopsis* contains 34 *CDPK* genes [26]. There are great differences in the number of *CDPK* genes in different species, including 21 in potato and 18 in melon [38,39]. In this study, 33 *ShCDPK* genes of *Solanum habrochaites* were identified by BLAST analysis. The physical and chemical properties of *ShCDPK* gene family members, such as gene length, amino acid, molecular weight, isoelectric point, and molecular weight, were determined by bioinformatics analysis. Through phylogenetic analysis of the gene family in *Arabidopsis* and pepper, 33 *ShCDPK* gene family members were divided into four subfamilies, which is consistent with the studies conducted in cucumber [40], pineapple [41], and grape [42], indicating that there are still some commonalities among different species.

CDPK was identified in pea and soybean for the first time. However, a subsequent study showed that CDPK can be found in green algae, oomycetes, and some protozoa, such as ciliates and capsicum. The *CDPK* gene family is more distributed on chromosomes 1, 10, and 11 and unevenly distributed on other chromosomes. The unbalanced distribution of genes may be related to species evolution and genetic variation. The phylogenetic tree constructed from *Solanum habrochaites*, *Arabidopsis*, and *Pepper* is divided into four subfamilies, and each subfamily has *ShCDPK* distribution. Group 1 included three pairs of vertical homologous pairs of *Solanum habrochaites* and *Pepper*. Groups 2 and 3 included two pairs of vertical homologous pairs between *Solanum habrochaites* and *Arabidopsis*. Group 4 did not contain vertical homologous pairs between *Solanum habrochaites* and *Arabidopsis*. These results showed that the homologous genes related to *Solanum habrochaites* and *Arabidopsis* had a conserved structure and function. In contrast, *Arabidopsis* and *Solanum habrochaites* have vertical homologous pairs, indicating that the *CDPK* gene family of *Solanum habrochaites* is relative to *Arabidopsis* in functional evolution. The *CDPK* gene family is mainly rich in introns, and the composition of intron exons can reflect the evolutionary relationship of the gene family. *ShCDPK8* and *ShCDPK12* present the most intron enrichment, containing 11 introns. *ShCDPK2, ShCDPK7*, and *ShCDPK9* have 10 introns. They are located in group 4 in the phylogenetic tree. Other introns were enriched in group 1, group 2, and group 3 of the phylogenetic tree. The number of introns in all gene family members was greater than 6, and there were no genes with intron deletion. The rate of intron acquisition is slower than that of intron deletion. The phylogeny of intron deleted *CDPK* gene may be the branch of intron enriched gene mRNA re inserted into the genome during the reverse transcription of intron enriched gene mRNA. There are 11 pairs of tandem repeat genes in *ShCDPKs*. The results showed that fragment duplication originated from stress. These 11 pairs of genes are highly consistent on the conserved motif, indicating that tandem replication plays a key role in the amplification of *ShCDPKs*. The KS of tandem repeat gene pairs is 0.0932819~1.30348, the separation time is 82~115.2 Mya, and it is 200~205 Mya during single dicotyledon differentiation. It is speculated that the fragment replication of *ShCDPKs* occurs after single dicotyledon differentiation [43].

It is well known that Ca^2+^ as a ubiquitous second messenger in the plant signal system plays a great role in plant growth and development. When plants are stimulated by environment and development, the stimulation triggered by external factors, such as temperature, light, salt, and osmotic stress, can produce different calcium ion changes. Specific calcium receptors can recognize and perceive these changes, and then play a role in regulating gene expression through a series of cascade reactions [44]. Through research, the *CDPK* gene senses the change of Ca^2+^ concentration through EF hand structure, relieves self-inhibition, activates kinase domain, and then transmits information to regulate the physiological changes of plants, widely participating in plant growth, development, and morphological construction [45]. In this study, the expression of *ShCDPK* family members changed significantly after cold, drought stress, and *Botrytis cinerea* stress. The expressions of *ShCDPK6, ShCDPK19,* and *ShCDPK26* were significantly up-regulated under low temperature stress, suggesting that these genes may be involved in cold response. Under cold stress, the expression of *ShCDPK12* and *ShCDPK30* decreased significantly, indicating that different members of CDPK may play different roles in response to cold stress. At the beginning of the drought and *Botrytis cinerea* treatments, the expression level of most genes increased significantly, the expression of *ShCDPK5, ShCDPK6, ShCDPK23, ShCDPK26, ShCDPK29, ShCDPK30,* and *ShCDPK33* genes increased significantly at 0.5 h of drought stress, and the expression of *ShCDPK6, ShCDPK16, ShCDPK23, ShCDPK26,* and *ShCDPK29* peaked at 0.5 h and 12 h. This may be because of the overrepresentation of cis acting elements relating to defense and drought stress in their promoter. The results show that CPK10 and HSP1 participate in the regulation of stomatal movement through ABA and Ca^2+^ signaling pathways, which may play an important role in the response of plants to drought stress [39]. It is also reported that guard cell anion channel SLAC1 is regulated by CPK21 and CPK23 and has different Ca^2+^ affinity [46]. AtCDPK4 and AtCDPK11 in *Arabidopsis* are responsive to salt stress. Grape VpCDPK16 was similar to AtCDPK4 and AtCDPK11, while they could phosphorylate ABA response transcription factors and reduce the tolerance to salt stress in seed germination and seedling growth. *VpCDPK16* also had similar expression [35]. In this study, under abiotic stress, such as low temperature and drought, and biotic stress of *Botrytis cinerea* infection, the expression of *ShCDPK6* and *ShCDPK26* changed significantly, indicating that these genes play an important role in the environmental adaptability of *Solanum habrochaites*. VIGS experiments were performed on *ShCDPK6* and *ShCDPK26*.The level of gene silencing is influenced by a variety of factors, including the size of the inserted silencing fragment, the direction of insertion, and the environment in which it is grown [47], and of course, genetic differences exist in plants of different species. All these factors may make the level of gene silencing different. This silencing system mediated by TRV (tobacco rattle virus) vector has been widely used in tomato varieties [48,49], but the presence of multiple factors leads to an average level of silencing. During biotic stress, the activities of SOD, POD, AAO, PPO, and CAT in silenced plants increased to varying degrees with stress stimulation, indicating that after the silencing of *ShCDPK6* and *ShCDPK26* genes, the enzyme activity response is more sensitive, so as to balance the metabolic system in order to eliminate the active oxygen produced by plant stress. From the expression of various enzyme activities, the enzyme activities of silenced plants are less than those of control plants, which may be caused by the silencing of *CDPK* family genes. In abiotic stress, we explored the changes of physiological indexes of silent plants under drought stress and cold stress. Under low temperature and drought stress, plants will produce a large amount of reactive oxygen species which can damage the plant cell membrane. In order to effectively remove reactive oxygen species from plant cells, plants activate a series of antioxidant systems, such as SOD and POD [50]. The results showed that there were some changes between the silent plants and the control plants under the two stresses. The change of POD in silent plants increased first and then decreased, but it was less than that in CK group, which may be because POD needs to remove hydrogen peroxide produced under stress and reduce the damage caused by stress. The change of SOD value increases first and then decreases as time goes on. However, compared with *ShCDPK6*, *ShCDPK26* gene may play a regulatory role under abiotic stress. These results further indicate that the lack of *ShCDPKs* family member genes make plant more sensitive to low temperature, drought, and *Botrytis cinerea*. However, how *ShCDPK6* and *ShCDPK26* respond to various stress conditions and functions needs to be further studied.

## 4. Materials and Methods

### 4.1. Plant Materials and Treatments

*Solanum habrochaites* (LA1777, https://tgrc.ucdavis.edu/, the National Plant Germplasm System (NPGS), California (for USA) accessed on 5 April 2022) was conserved in our lab supplying by the Tomato Genetics Research Center (University of California, Davis, CA, USA). The seeds were first soaked in 2% sodium hypochlorite solution for 16 min (slowly shaken), then rinsed with sterile water 3–4 times, then soaked in 75% alcohol for 30 s, and rinsed with sterile water. Sterilized seeds were pre-germinated on wet filter paper and subsequently transferred to plastic basin of the soil mixture (1 V:1 V). They were planted in an artificial climate chamber at 21–26 °C, 16 h/8 h (day/night). Tomato seedlings at five leaves and one heart were used for subsequent experimental treatment. Three plants with the same growth were taken from each treatment for three biological replicates. Three technical replicates were performed for each treatment.

#### 4.1.1. Cold Stress

Plant roots were dipped in half-strength Hoagland culture solution for 2–3 days. Cold stress was exerted by exposing a subset of the plants to low temperature (4 °C). Samples were obtained at time points 0, 0.5, 3, 9, 12, 24, and 36 h.

#### 4.1.2. Drought Stress

Roots of experimental and control plants were dipped in half-strength Hoagland culture solution for 2–3 days. The treatment group and the control group were sampled simultaneously. Treatment group: The roots of plants were soaked with 20% PEG6000 (1/4 Hoagland nutrient solution). They were allowed to continue growing in this medium until samples were obtained. Samples were obtained at time points 0, 0.5, 3, 9, 12, and 24 h.

#### 4.1.3. Botrytis Cinerea Stress

*B. cinerea* spores were diluted with sterile water to make the concentration of spore suspension 10^7^ CFU/mL. Treatment group: The fungus solution was sprayed on the tomato leaves and cultured in a light incubator with 90% humidity at 20–25 °C, while in the control group the tomato blade was sprayed with water, and other conditions remained constant. Samples were obtained at time points 0, 0.5, 3, 9, 12, 24, and 36 h.

### 4.2. Identification of the CDPK Gene Family in Solanum habrochaites by Bioinformatic Analysis

#### 4.2.1. Identification and Basic Information Analysis of *CDPK* Gene Family Members in *Solanum habrochaites*

Firstly, the hidden Markov model (HMM) files corresponding to protease (PF00069) and NaF domain (PF03822) were downloaded from Pfam protein family database (https://pfam.xfam.org/ accessed on 24 July 2020). Then, the whole genome data of *Solanum habrochaites* were obtained by hmmsearch (https://www.ncbi.nlm.nih.gov/data-hub/taxonomy/62890/ accessed on 24 July 2020). CDPK gene (E-value < 1 × 10^−7^, similarity >50%), and the members and number of *CDPK* gene family in *Solanum habrochaites* were obtained. The length, molecular weight, and isoelectric point of CDPK protein in *Solanum habrochaites* were predicted by bio-linux software. In order to understand the chromosome situation of the gene family, the chromosome mapping map was drawn by MapChart software. At the same time, SOSUI (http://harrier.nagahama-i-bio.Ac.jp/sosui/ accessed on 24 July 2020) online software was used to predict the protein transmembrane domain. CELLO (http//cello.life.nctu.edu.tw accessed on 24 July 2020) was used to predict subcellular localization and identify the functional sites of the *CDPK* gene family.

#### 4.2.2. Evolutionary Analysis of *CDPK* Gene Family in *Solanum habrochaites*

In order to explore the evolutionary relationship between the *CDPK* gene family, comparative analysis of the *CDPK* gene family in *Solanum habrochaites*, *Arabidopsis,* and peppers was conducted using MEGA software with bootstrap repeated 1000 times.

#### 4.2.3. Analysis of Genetic Structure of *CDPK* Gene in *Solanum habrochaites*

Genomic sequence of *CDPK* genes consisting of the coding sequence and the non-coding regions were obtained for *Solanum habrochaites*. The GSDS tool (http://gsds.cbi.Pku.Edu.cn/ accessed on 24 July 2020) was used to perform gene structure analysis and draw the exon-intron structure diagram.

#### 4.2.4. Analysis of *CDPK* Gene Family Conservative Motif in *Solanum habrochaites*

In order to study the evolutionary relationship of *CDPK* gene family in *Solanum habrochaites*, 10 conserved motifs of *CDPK* genes were analyzed online with MEME (https://meme-suite.org/meme/ accessed on 24 July 2020).

#### 4.2.5. Gene Replication of *CDPK* Gene Family in *Solanum habrochaites*

The full-length amino acid sequence is compared using Clustal. Ka is the non-synonymous substitution rate, and Ks is the synonymous substitution rate. Ks and Ka programs were calculated using Kaks_Calculator 2.0 and the discrete time of genetic pairs was estimated using mutation rate. The substitution of each synonymation in each year is T = Ks/2X (x = 6.56 × 10^−9^). If Ka/Ks > 1, a positive selection effect is considered. If Ka/Ks = 1, neutral selection is considered to exist. If Ka/Ks < 1, a purifying selection effect is considered.

### 4.3. Real-Time Quantitative PCR (qRT-PCR) Analysis

The plants were grown at 25 °C for 4 weeks. The total RNA of samples were extracted using Trizol reagent (Invitrogen, California (for USA)) and used for mRNA synthesis with High-Capacity cDNA Reverse Transcription Kits (ThermoFisher Scientific, Waltham, MA, USA) following the manufacturer’s recommendations. qRT-PCR was performed with the SYBR Green qPCR Mix Kit (Biosharp, Hefei, China). NCBI (https://www.ncbi.nlm.nih.gov/ accessed on 15 July 2021) Primer Blast was used to design fluorescent quantitative primers (MAX value to 200 base pairs), and the primer sequence is shown in Appendix A. The product cDNA was diluted 10 times and used as template for qRT-PCR. Each reaction contained 10 µL Chamq SYBR qPCR Master Mix (Without Rox) and 0.4 µL forward primer (10 μM), 0.4 µL reverse primer (10 μM), and cDNA samples diluted 10 times with 1.0 µL. PCR amplification was performed using a Quant Studio real-time fluorescent quantitative PCR instrument. Reaction conditions: predeformation, 95 °C for 5 min, followed by 40 cycles, 95 °C for 10 s, 60 °C for 30 s. Data represent means of three replicates ± standard deviation (SD). Analysis was performed using the Data Processing System, and Tukey’s multiple range test was conducted to determine significant differences. *p* < 0.05 was considered to indicate statistical significance.

### 4.4. Virus-Induced Gene Silencing (VIGS)

The successfully constructed *pTRV2-ShCDPK6* and *pTRV2-ShCDPK26* were transformed into *Agrobacterium* *tumefaciens* strain GV3101 together with pTRV1 and *pTRV2-PDS*. Cell culture containing the appropriate plasmid was injected into the tomato seedlings through the leaves with a syringe. *A**grobacterium* *tumefaciens* containing *PTRV1* was mixed with *agrobacterium* *tumefaciens* containing *pTRV2-**S**hCDPK6*, *pTRV2-**S**hCDPK26* and *pTRV2-PDS* respectively at 1:1 and placed at room temperature for 3 h. Plant leaves were inoculated with a 1 mL syringe to fill the injected leaves with bacterial solution to the greatest extent (the leaves are not damaged). Hence, 50 seedlings were infected at one time, among which 5 seedlings were used as the *pTRV2-PDS* plants,15 seedlings used for the drought treatment, 15 seedlings used for cold treatment, and 15 seedlings used for the *Botrytis cinerea* treatment. Each plant was inoculated 3 times, with 1 mL each time. At the same time, the *p**TRV2-PDS* empty vector was inoculated into plant leaves as the positive control group. The soaked tomatoes were incubated overnight in dark conditions at 22 °C, and then incubated in a constant temperature incubator with a light cycle of 16 h/8 h. Other conditions were not changed. To ensure successful infiltration, tomato plant materials were inoculated with different combinations of fluid in each 4 min vacuum process. After the positive control plants showed the phenomenon of melatonin, the leaves of the experimental group and the control group were taken to detect efficiency of silencing gene using qRT-PCR [51]. The experiment was repeated three times. After the target fragment was successfully silenced, five physiological indexes of superoxide dismutase (SOD), catalase (CAT), ascorbate oxidase (AAO), peroxidase (POD), and polyphenol oxidase (PPO) were determined.

## 5. Conclusions

In this study, 33 *CDPK* gene family members of *Solanum habrochaites* were identified, which can be divided into four subgroups, mainly distributed in the cytoplasm with a few in the outer membrane. Phylogenetic analysis showed that the *CDPK* gene family had similar genes between *Solanum habrochaites* and cultivated tomato. Structural analysis showed that there was intron enrichment in the *CDPK* family of *Solanum habrochaites*. Motif analysis showed that the gene motifs with high homology of the *CDPK* gene family in *Solanum habrochaites* had strong similarity. According to the analysis of cis acting elements, the *CDPK* gene family of *Solanum habrochaites* is related to defense and stress response, anaerobic induction, drought stress, and other functions. The results of qRT-PCR showed that the *CDPK* gene family could respond to low temperature, drought, and *Botrytis cinerea* stress, and *ShCDPK6* and *ShCDPK26* were significantly expressed under the three stresses. The stress resistance of *ShCDPK6* and *ShCDPK26* silenced plants decreased significantly, indicating that ShCDPK6 and ShCDPK26 play an important role in response to abiotic and biotic stress.

## Figures and Tables

**Figure 1 ijms-23-04227-f001:**
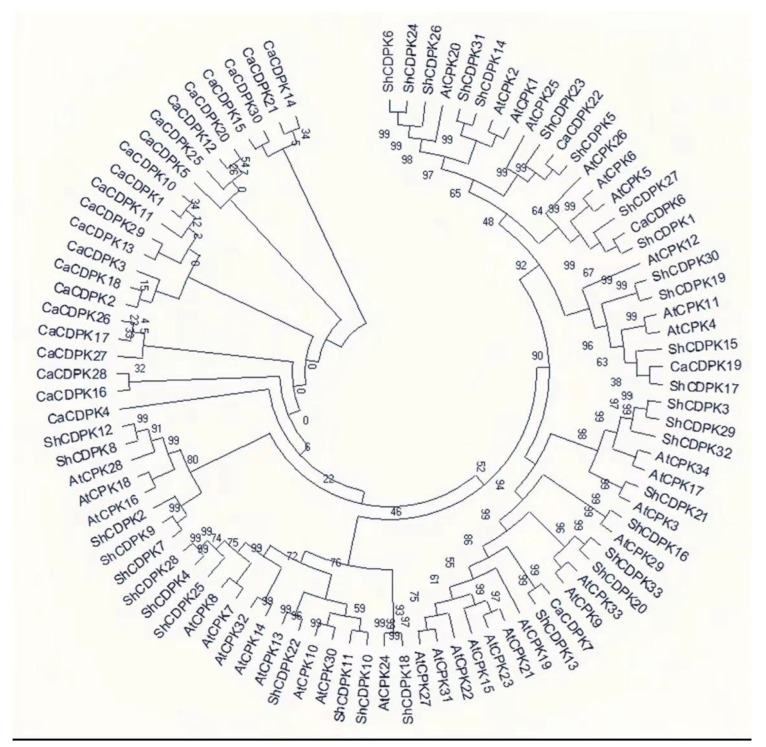
Phylogenetic tree analysis of *CDPK* gene family in *Solanum habrochaites*.

**Figure 2 ijms-23-04227-f002:**
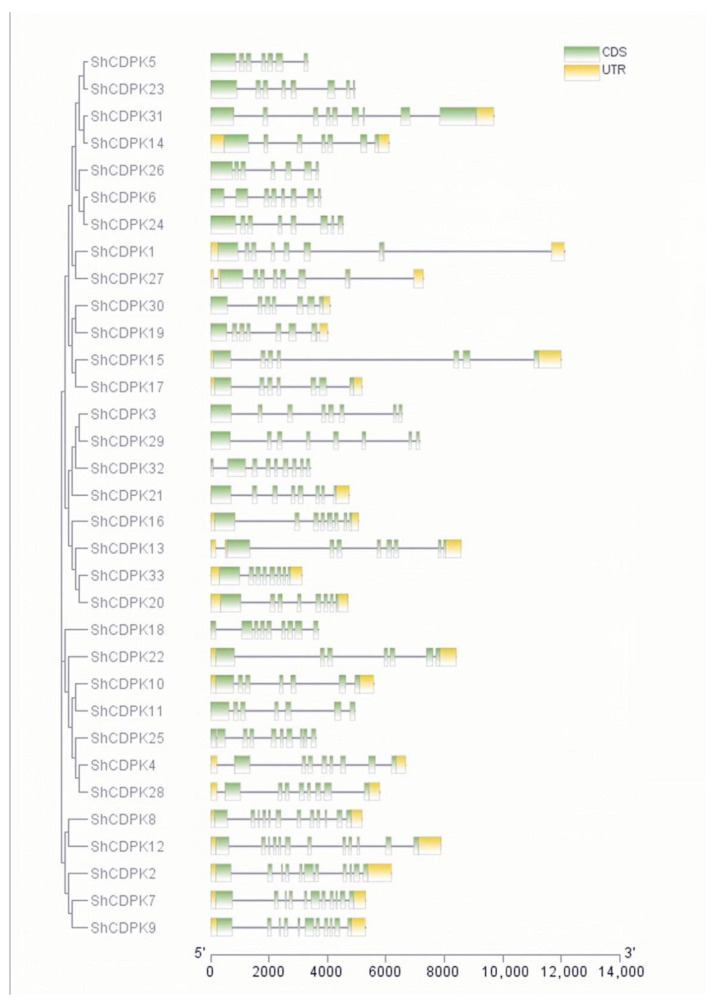
Exon and intron structure analysis of *CDPK* gene family in *Solanum habrochaites*.

**Figure 3 ijms-23-04227-f003:**
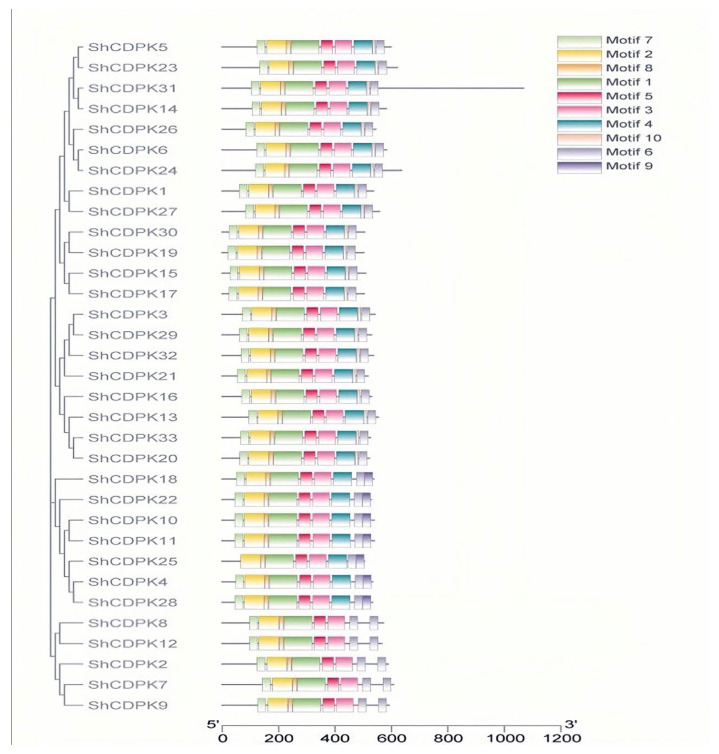
Conserved motif types of *CDPK* gene family in *Solanum habrochaites*.

**Figure 4 ijms-23-04227-f004:**
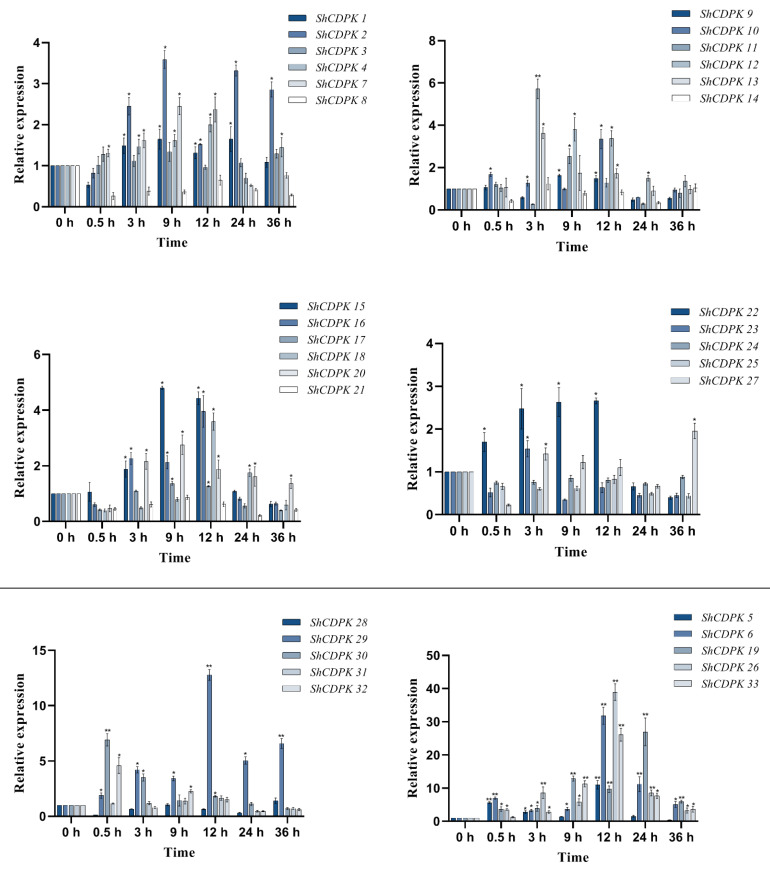
Expression analysis of *CDPKs* gene in *Solanum habrochaites* with real-time PCR under cold stress. Quantitative PCR data are represented as mean ± SEM. ** *p* < 0.01, * *p* < 0.05. All date represent mean values for three independent biological replicates.

**Figure 5 ijms-23-04227-f005:**
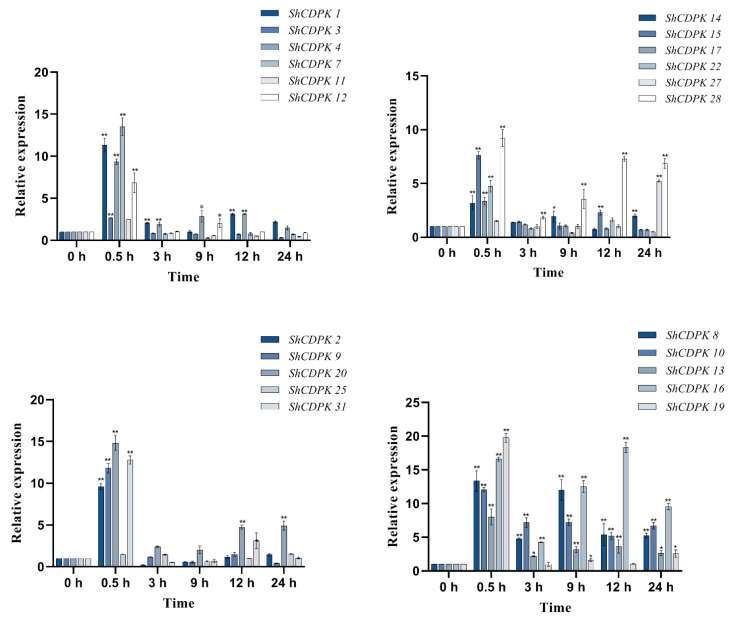
Expression analysis of *CDPKs* gene in *Solanum habrochaites* with real-time PCR under drought stress. Quantitative PCR data are represented as mean ± SEM. ** *p* < 0.01, * *p* < 0.05. All date represent mean values for three independent biological replicates.

**Figure 6 ijms-23-04227-f006:**
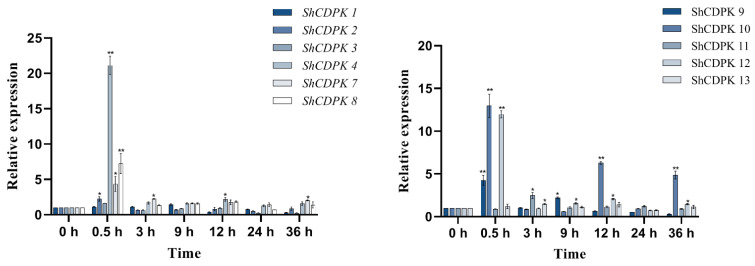
Expression analysis of *CDPKs* gene in *Solanum habrochaites* with real-time PCR under *Botrytis cinerea* stress. Quantitative PCR data are represented as mean ± SEM. ** *p* < 0.01, * *p* < 0.05. All date represent mean values for three independent biological replicates.

**Figure 7 ijms-23-04227-f007:**
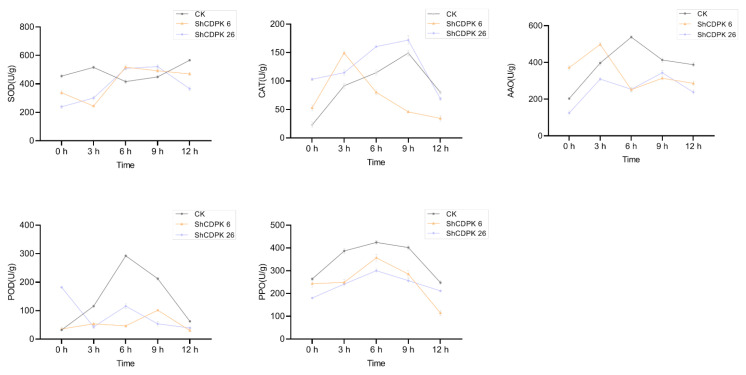
Determination of physiological indexes of *ShCDPK6* and *ShCDPK26* silenced plants and control plants under cold stress. Activity of SOD, CAT, AAO, POD, PPO. Each value is the mean ± standard deviation of at least three independent measurements.

**Figure 8 ijms-23-04227-f008:**
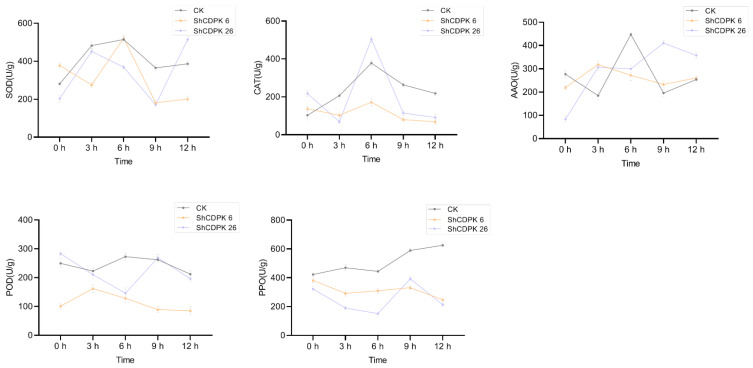
Determination of physiological indexes of *ShCDPK6* and *ShCDPK26* silenced plants and control plants under drought stress. Activity of SOD, CAT, AAO, POD, PPO. Each value is the mean ± standard deviation of at least three independent measurements.

**Figure 9 ijms-23-04227-f009:**
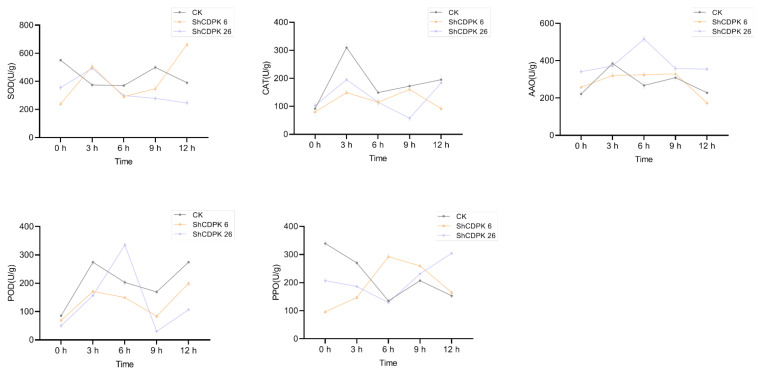
Determination of physiological indexes of *ShCDPK6* and *ShCDPK26* silenced plants and control plants under *Botrytis cinerea* stress. Activity of SOD, CAT, AAO, POD, PPO. Each value is the mean ± standard deviation of at least three independent measurements.

**Table 1 ijms-23-04227-t001:** Characteristic of *CDPK* gene family in *Solanum habrochaites*.

Gene Name	Sequence Accession	Gene Length (bp)	Protein Length (aa)	Mv/Da (ku)	PI	Chromosome	Chromosome Starting Position	Chromosome Termination Position	Subcellula Localization
*ShCDPK1*	Solhab01g001000	1608	535	60,015.8	5.8	Chr1	153266	165357	1
*ShCDPK2*	Solhab01g038800	1767	588	65,836	9.25	Chr1	2898264	2904451	1
*ShCDPK3*	Solhab01g408500	1626	541	61,077.6	5.52	Chr1	101718341	101724893	1
*ShCDPK4*	Solhab01g412600	1602	533	60,048.3	7.28	Chr1	102121744	102128426	1
*ShCDPK5*	Solhab01g424200	1797	598	67,570.6	5.22	Chr1	103438229	103441551	1
*ShCDPK6*	Solhab01g425400	1749	582	64,677	5.93	Chr1	103522036	103525799	1
*ShCDPK*7	Solhab02g041900	1824	607	67,931.8	9.51	Chr2	3447317	3452629	1
*ShCDPK*8	Solhab02g106200	1713	570	64,271.5	9.77	Chr2	8674949	8680139	1.2
*ShCDPK*9	Solhab02g234000	1776	591	66,246.1	8.72	Chr2	19980037	19985317	1
*ShCDPK*10	Solhab03g087000	1617	538	60,922.5	6.85	Chr3	7056576	7062155	1
*ShCDPK*11	Solhab03g089800	1620	539	60,876.4	6.98	Chr3	7284876	7289824	1
*ShCDPK*12	Solhab03g289900	1698	565	63,959	9.39	Chr3	67058383	67066282	1.2
*ShCDPK*13	Solhab03g297700	1662	553	62,883.1	6.84	Chr3	67914192	67922776	1
*ShCDPK*14	Solhab04g006300	1746	581	64,561.9	5.56	Chr4	747800	753899	1
*ShCDPK*15	Solhab04g100300	1527	508	57,230.7	4.88	Chr4	20036567	20048558	1
*ShCDPK*16	Solhab04g295200	1590	529	59,692.7	5.17	Chr4	69050936	69056009	1
*ShCDPK*17	Solhab05g000900	1512	503	56,429.8	4.77	Chr5	201594	206779	1
*ShCDPK*18	Solhab06g061000	1611	536	61,044.4	6.02	Chr6	4514265	4517937	1
*ShCDPK19*	Solhab06g118900	1506	501	56,383.9	5.78	Chr6	9089226	9093227	1
*ShCDPK20*	Solhab07g020200	1566	521	57,818.7	7.05	Chr7	1368173	1372887	1
*ShCDPK21*	Solhab08g229700	1551	516	57,751	6.02	Chr8	65522651	65527398	1
*ShCDPK22*	Solhab09g271700	1590	529	59,620.4	6.39	Chr9	85410171	85418586	1
*ShCDPK23*	Solhab10g042100	1863	620	69,779.1	5.57	Chr10	3099146	3104072	1
*ShCDPK24*	Solhab10g043000	1908	635	69,576.7	5	Chr10	3180473	3185006	1
*ShCDPK25*	Solhab10g069000	1515	504	57,051.6	6.06	Chr10	5261738	5265333	1
*ShCDPK26*	Solhab10g082100	1635	544	60,419.3	5.44	Chr10	6302023	6305715	1
*ShCDPK27*	Solhab10g101400	1674	557	62,198.8	5.59	Chr10	8753228	8760508	1
*ShCDPK28*	Solhab11g059000	1602	533	59,654.8	6.41	Chr11	5422926	5428709	1
*ShCDPK29*	Solhab11g068700	1590	529	59,533	5.97	Chr11	6787516	6794660	1
*ShCDPK30*	Solhab11g157600	1518	505	56,832.4	5.4	Chr11	56529976	56534049	1
*ShCDPK31*	Solhab11g234700	3204	1067	120,659.3	7.4	Chr11	65019137	65028821	1
*ShCDPK32*	Solhab12g006400	1608	535	59,607.1	5.31	Chr12	430517	433905	1
*ShCDPK33*	Solhab12g260400	1578	525	58,321	6.79	Chr12	69390749	69393879	1

Note: Subcellular localization: 1. Cytoplasmic, 2. Outer membrane.

## Data Availability

Not applicable.

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
