# Peer review of "Identification of CDPK Gene Family in Solanum habrochaites and Its Function Analysis under Stress"

_ijms, 2022, doi:10.3390/ijms23084227_

Round 1

Reviewer 1 Report

  1. At the end of the INTRODUCTION chapter, I ask the authors to write more clearly the hypotheses of the experiences.
  2. R 242. In many parts of the world Phytoptora infestans is the most dangerous disease in tomatoes, not Botrytis cinerea.
  3. R ​​441. From which organs were spores taken for artificial infections with Botrytis cinerea?
  4. R. 288-289. In order to evaluate the factors of biotic and abiotic stress (Botrytis cinerea, drought and cold), apart from the evaluation of the degree of oxidative damage, were macroscopic observations made in field conditions, on plants? If such observations were made, what was the procedure used? I consider this stage of evaluation to be necessary and important. Finally, all the results of molecular biology research must be found in concrete, measurable results in the field.
  5. R 442 Do the authors refer to artificial infections with Botrytis cinerea using bacterial solution? I mention that Botrytis cinerea is a fungus, not a bacterium.

Author Response

Thank you for your suggestions. All your suggestions are very important. They are of great guiding significance to my thesis writing and scientific research.

The following summarizes how we responded to reviewer comments.

Referee #1

At the end of the INTRODUCTION chapter, I ask the authors to write more clearly the hypotheses of the experiences.

Answer: Thank you for reminding us that we have modified the problems. We have added the hypotheses in the article.

R 242. In many parts of the world Phytoptora infestans is the most dangerous disease in tomatoes, not Botrytis cinerea.

Answer: Thank you for your suggestions that we have modified this issue in Line 252.

R 441. From which organs were spores taken for artificial infections with Botrytis cinerea?

Answer: Thank you for your suggestions that we have modified this issue in Line 463.

  1. 288-289. In order to evaluate the factors of biotic and abiotic stress (Botrytis cinerea, drought and cold), apart from the evaluation of the degree of oxidative damage, were macroscopic observations made in field conditions, on plants? If such observations were made, what was the procedure used? I consider this stage of evaluation to be necessary and important. Finally, all the results of molecular biology research must be found in concrete, measurable results in the field.

Answer: Thank you very much for your question, we have not made observations in field conditions yet, and we will consider applying it to a large field trial in the future. However, we observed the phenotype of silent plants under low temperature, drought and Botrytis cinerea stress. The specific results are shown in supplementary Figure 3. Compared with the control plants, the silent plants showed more wilting under low temperature and drought stress, but the phenotypic difference was not very obvious under Botrytis cinerea stress. These results will be the basis for large-scale field experimental research in the future.

R 442 Do the authors refer to artificial infections with Botrytis cinerea using bacterial solution? I mention that Botrytis cinerea is a fungus, not a bacterium.

Answer: Thank you for your suggestions that we have modified this issue in Line 463.

Reviewer 2 Report

The authors provide a genome wide analysis of the CDPK gene family in hairy tomato Solanum habrochaites, together with gene expression analysis under various biotic and abiotic stress, as well as functional characterization using 2 VIGS lines.

What are the primer sequences and more importantly the targeted regions of ShCDPK6 and ShCDPK26? These two genes are very closed, how did you succeed in targeting only one of these two genes on each silenced plant? Same remark for ShCDPK24. Are you sure that ShCDPK24 was not able to "complement" the functional loss for each line? Provided at least gene expression for all these 3 genes for each line.

How many independent lines analyzed for the silenced plants?

The obtained silenced level is not high. Are very distinct plants, in terms of gene expression silenced level considered? Not sure since the error bars are very small. So, do you explain this very low efficiency? Is this type of VIGS well adapted to hairy tomato? Please provide a reference or explain how you have verified it?

Provide the primer sequences for the reference genes used for RT-qPCR analyses.

Provide each hybridization temperature for each primer couples (sequences are very different, so the same temperature must be different), and melting temperature for each CDPK amplified sequence obtained by RT-qPCR.

Interpretation of Ka/Ks ratio has to be completed and revised.

Agrobacterium tumefaciens, not Agrobacterium

 Plant, bacteria and fungi species in italics (Arabidopsis thaliana, Solanum habrochaites, Agrobacterium tumefaciens, Botrytis cinerea)

Author Response

Thank you for your suggestions. All your suggestions are very important. They are of great guiding significance to my thesis writing and scientific research.

The following summarizes how we responded to reviewer comments.

Referee #2:

The authors provide a genome wide analysis of the CDPK gene family in hairy tomato Solanum habrochaites, together with gene expression analysis under various biotic and abiotic stress, as well as functional characterization using 2 VIGS lines.

What are the primer sequences and more importantly the targeted regions of ShCDPK6 and ShCDPK26? These two genes are very closed, how did you succeed in targeting only one of these two genes on each silenced plant? Same remark for ShCDPK24. Are you sure that ShCDPK24 was not able to "complement" the functional loss for each line? Provided at least gene expression for all these 3 genes for each line.

Answer: Thank you very much for your suggestion. ShCDPK6 and ShCDPK26 gene sequences were used https://vigs.solgenomics.net/. The silenced fragments of ShCDPK6 and ShCDPK26 genes were obtained as the target regions of the two genes. We designed specific primers for the silenced fragments and connected them to the silencing expression vector, which can effectively silence ShCDPK6 and ShCDPK26 respectively. In bioinformatics, ShCDPK24 gene in this gene family was found to have gene replication and covariance with ShCDPK6 and ShCDPK26. After gene sequence matching, it was found that the silenced fragments of these three genes were consistent, but there was still 38.4% variation (sequence matching) to a certain extent. Therefore, after synthesizing specific silencing fragments, we believe that specific genes can be silenced. Meanwhile, we detected the expression changes of ShCDPK24 gene under ShCDPK6 and ShCDPK26 gene silencing (Figure S2), but the expression of ShCDPK24 gene in the two silenced plants was not significant, while the expression of shcdpk6 and ShCDPK26 genes decreased more significantly, as shown in Figure S2. Therefore, we speculate that although ShCDPK24 has a close evolutionary relationship with ShCDPK6 and ShCDPK26 from the perspective of bioinformatics analysis, the expression level of this gene does not show a "complementary" function with other genes in silent ShCDPK6 and ShCDPK26 plants. 

How many independent lines analyzed for the silenced plants?

Answer: Thank you for reminding us that we have analyzed 15 independent strains for each of the two silenced plants.

The obtained silenced level is not high. Are very distinct plants, in terms of gene expression silenced level considered? Not sure since the error bars are very small. So, do you explain this very low efficiency? Is this type of VIGS well adapted to hairy tomato? Please provide a reference or explain how you have verified it?

Answer: Thank you very much for your comments, we think this is an important issue. We have revised this problem in line 415-425 and added some references.

Provide the primer sequences for the reference genes used for RT-qPCR analyses.

Answer: Thank you for reminding us that we have modified the problems. We have added the primer sequences in Table S1.

Provide each hybridization temperature for each primer couples (sequences are very different, so the same temperature must be different), and melting temperature for each CDPK amplified sequence obtained by RT-qPCR.

Answer: Thank you for reminding us that we have modified the problems. We have added on the solubilization temperature of primers in Table S1.

Interpretation of Ka/Ks ratio has to be completed and revised.

Answer: Thank you for reminding us that we have modified the problems. We have completed the revision of Ka/Ks ratio in 4.2.5.

Agrobacterium tumefaciens, not Agrobacterium. 

Answer: Thank you for your suggestions that we have changed “Agrobacterium” to “Agrobacterium tumefaciens” in the full text.

Plant, bacteria and fungi species in italics (Arabidopsis thaliana, Solanum habrochaites, Agrobacterium tumefaciens, Botrytis cinerea)

Answer: Thank you for your advice. All your suggestions are important. We have revised this problem in the full text. We checked the full text and revised the plant, bacteria and fungi species in italics.

Round 2

Reviewer 2 Report

The Authors have considered and replied to all my queries.